# Pulmonary Parameters in Adolescents with Severe Thoracic Idiopathic Scoliosis: Comparison Girls versus Boys

**DOI:** 10.3390/healthcare10081574

**Published:** 2022-08-19

**Authors:** Katarzyna Politarczyk, Wiktoria Popowicz-Mieloch, Tomasz Kotwicki

**Affiliations:** 1Department of Spine Disorders and Pediatric Orthopaedics, University of Medical Sciences, 61-545 Poznan, Poland; 2Department of Preventive Health, University of Medical Sciences, 60-781 Poznan, Poland

**Keywords:** idiopathic scoliosis, pulmonary function test, body height

## Abstract

The study compared pulmonary parameters, registered at the preoperative examination, in adolescent boys versus girls, both with severe thoracic idiopathic scoliosis. Thirty consecutive boys and 30 consecutive girls with Lenke 1 or 3 type, in the age range 14–18 years, with a Cobb angle of >50° and Risser sign ≥ 3 were enrolled. Corrected body height was used to calculate pulmonary parameters according to the Global Lung Function (GLI 2012) reference values. Significantly higher values of the calculated predicted pulmonary parameters and the upper and lower limit of normal (ULN and LLN), as well as significantly higher absolute values of forced vital capacity (FVC) and forced expiratory volume in 1 s (FEV1), were observed in boys than girls; however, the registered FVC and FEV1, expressed as percentages of the predicted values, tended to be lower in boys. The FEV1 z-score difference between boys and girls may suggest a need for more intensive preoperative pulmonary rehabilitation in boys.

## 1. Introduction

Severe thoracic idiopathic scoliosis is more rarely observed in boys than in girls [1,2]. Several studies comparing boys and girls concerning growth and maturation [3], the risk of scoliosis progression [4,5], the brace treatment outcomes [6,7] or surgical treatment outcomes [8,9] have been published. A negative relationship between severe Cobb angle, loss of thoracic kyphosis or increased number of vertebrae involved in the curvature and the pulmonary parameters has been previously described in adolescents with idiopathic scoliosis [10,11,12,13,14].

Johnston et al. [10] reported that absolute forced vital capacity (FVC) values and forced expiratory volumes in 1 s (FEV1) values were significantly higher in boys than in girls (*p* < 0.0001), while the percentages of the predicted values of the pulmonary parameters did not reveal a significant difference between genders. However, the authors did not consider upper limit of normal (ULN), the lower limit of normal (LLN) or the z-score values.

The aim of the study was to compare the values of the pulmonary parameters obtained at preoperative spirometry examination in adolescent boys versus girls, both having severe thoracic idiopathic scoliosis.

## 2. Materials and Methods

### 2.1. Study Population

The hospital database was queried to reveal boys who underwent surgical treatment for idiopathic scoliosis and met the inclusion criteria: Lenke 1 or 3 type, Risser sign ≥ 3, aged 14–18 years old, preoperative spirometry examination, complete documentation. Thirty consecutive boys were enrolled. Then, 30 consecutive girls who met the same inclusion criteria were included.

Retrospective analysis of preoperative radiographs and preoperative spirometry examination results was performed. Patients’ body height, body weight, and the absolute values of the FVC and FEV1 were measured.

### 2.2. Radiological Examination

The Cobb method was used to measure the value of the thoracic curve on anteroposterior standing radiograph of the whole spine [15]. The bone maturity was assessed based on the Risser sign [16].

### 2.3. Corrected Body Height Calculation

Stokes’ formula was used to calculate the loss of body height induced by the spinal curvature. In case of a single curve, the following formula was used to calculate the loss of the body height: 1.55 − 0.0471 Cobb + 0.009 Cobb^2^, while, for the double curve, the formula 1.0 + 0.066 Cobb + 0.0084 Cobb^2^ was used [17]. The sum of the loss of body height and the measured body height equals the corrected body height [18].

The corrected body height was taken instead of the measured body height as the base for calculations of the predicted pulmonary parameters, the LLN, the ULN, the percentages of the predicted values of the parameters and z-score [19].

### 2.4. Pulmonary Parameters Calculation

The Global Lung Function Initiative (GLI 2012) software for calculating the predicted values for individuals (available to download on the European Respiratory Society website) was used to calculate the predicted values of the pulmonary parameters (FVC and FEV1), the lower limit of normal (LLN), the upper limit of normal (ULN) and the percentages of the predicted values of the pulmonary parameters, as well as the z-score values. The Global Lung Function Initiative reference values were applied throughout the study [20].

### 2.5. Pulmonary Testing

The spirometry examination was performed in sitting position with corrected body posture using a LungTest LT 250 spirometer (MES, Kraków, Poland). The appropriate technique was described and explained to the patients before performing the spirometry examination. All patients performed a minimum of 3 maneuvers of maximal inhalation followed by maximal exhalation until no more air could be expelled. Throughout the examination, enthusiastic coaching was performed to encourage patients to give maximal effort. Out of the obtained results, the best record of the FVC and FEV1 was taken for analysis [13,19,21,22,23,24].

### 2.6. Statistical Analysis

The Shapiro–Wilk test was used to analyze data distribution. Since the data revealed normal distribution, the Student’s *t*-test was used to determine significance of the difference between the pulmonary parameters. The age, measured body height, the corrected body height and body weight were compared between genders using the Student’s *t*-test. For a comparison of the Risser sign, the Wilcoxon test was performed. The analysis was performed with IBM SPSS Statistics 26 (New York, NY, USA). The statistical significance level was set at *p* = 0.05.

## 3. Results

The demographic data are presented in Table 1.

The loss of body height was similar in boys and girls (mean 3.95 cm vs. 3.56 cm, respectively, *p* = 0.297). In both genders, the values of the measured body height were significantly lower (*p* < 0.001) than the values of the corrected body height.

### 3.1. Radiological Parameters

No significant difference between the thoracic Cobb angle in boys versus girls (69.6° vs. 64.9°, respectively, *p* = 0.172) was found. Additionally, the median of the Risser sign did not differ in boys versus girls (4.0 vs. 4.0, respectively, *p* = 0.717).

### 3.2. Predicted Pulmonary Parameters Calculated for Boys versus Girls

The calculations revealed significantly higher absolute values of the predicted pulmonary parameters (FVC and FEV1), as well as the LLN and the ULN, for boys than for girls (*p* < 0.01), as shown in Table 2.

### 3.3. Pulmonary Parameters Registered during Spirometry Examination

The FVC and FEV1 absolute values were significantly higher in boys in comparison to girls, shown in Table 3.

The values of the z-score calculated for the FVC and FEV1 parameters measured at the spirometry examination and the values of the %FVC and %FEV1 were lower in boys than in girls; the difference was not significant (Table 4).

## 4. Discussion

Adolescent idiopathic scoliosis is observed in 2–3% of the general population. The prevalence is similar in boys and girls at the early stage when the Cobb angle values are within 10–20° range (1:1.3). However, the greater the Cobb angle value, the higher the risk of progression in girls than in boys—with a Cobb angle of more than 30°, the male-to-female risk ratio increases to 7:1. Out of the diagnosed cases, approximately 0.1–0.3% require surgical treatment [25,26].

The aim of this study was to compare the values of the pulmonary parameters in boys versus girls with severe thoracic idiopathic scoliosis. The study group consisted of 60 patients (30 boys and 30 girls) with the major curve localized in the thoracic region (Lenke 1 or Lenke 3 type), who did not differ for age, Risser sign or the Cobb angle. Boys were revealed to be significantly heavier and taller, considering both the values of the measured body height and the corrected body height. The loss of the body height calculated based on Stokes’ formula did not differ in boys versus girls. The predicted values of the pulmonary parameters, as well as the ULN and the LLN values, were calculated according to the GLI 2012 reference values endorsed based on the gender, age, ethnicity and body height [20]. As a substitute for the measured body height, which may be decreased as a consequence of the spine deformity, the corrected body height was used. In our previous studies [18,27], the difference between the measured and the corrected body height was found to be significant (*p* < 0.001). In addition, using the corrected body height instead of the measured body height significantly changed the calculation of the predicted values of the pulmonary parameters and influenced the interpretation of the pulmonary status of the patients with severe thoracic idiopathic scoliosis [18,27].

The absolute values of the FVC and FEV1 achieved during spirometry examination were revealed to be significantly higher in male than in female patients (*p* < 0.001 and *p* = 0.036, respectively). On the other hand, when the absolute values of both pulmonary parameters were recalculated into percentages of the predicted values (%FVC and %FEV1), no significant male versus female difference was found (%FVC: 70.24 vs. 73.19, *p* = 0.776; %FEV1: 70.35 vs. 75.83, *p* = 0.414). These findings are consistent with Johnston et al.’s [10] results, which concluded that the absolute FVC and FEV1 values were significantly higher in boys that in girls (*p* < 0.0001), while the percentages of the predicted values of FVC and FEV1 showed no significant difference between genders.

According to the American Thoracic Society and European Respiratory Society, the fifth percentile is recommended as an LLN that equals the z-score of −1.64 [28,29,30]. The z-score is not biased by the age, gender or ethnicity and seems to be more useful for defining the LLN value in contrast to the percentages of the predicted values of the pulmonary parameters [20]. In both boys and girls, the z-score values indicate that the FVC and FEV1 values obtained during spirometry examination were lower than expected. The z-score values tended to decrease in boys compared to girls (FVC z-score: −2.44 vs. −2.29, *p* = 0.701; FEV1 z-score: −2.52 vs. −1.99, *p* = 0.262). The results suggest pulmonary impairment in both genders. Although the FEV1 z-score difference was not statistically significant in boys versus girls, the value of 0.53 may indicate that the difference was clinically significant. According to the Global Lung Function advisory panel consensus, a z-score difference of >0.5 is considered to be clinically significant and is equated to a change of the percentage of the predicted value of approximately 6% [31].

Previous studies confirmed that, among healthy individuals who have the same height and weight, males have larger lungs volume than females and, as a consequence, a larger number of the bronchi, greater alveolar surface area and wider caliber of the airways [32]. However, the number of alveoli per unit of the surface area is the same in boys and girls. Later on, during adolescence, the differential growth between airway size and lung size occurs, called dysanapsis. As its’ consequence, in males, the disproportionate airway growth results in a much smaller number of alveoli compared to the number of airways, while, in females, airways growth is proportional to the lung tissue growth. Males have longer conducting airways than females, which is disadvantageous during this period of life [33]. In patients with severe thoracic idiopathic scoliosis, the spine and the chest deformation may impact the lung function. As studies suggest, in infantile scoliosis the alveoli multiplication is decreased, whereas, in juvenile or adolescent scoliosis, the alveoli may not enlarge normally. In addition, proportional to the lung maldevelopment, the number of the pulmonary vessels can be reduced [34].

According to Leong et al. [35], who studied three-dimensional kinematics of the chest and the spine during breathing in 41 patients with scoliosis and 20 non-scoliotic individuals, the range of motion of the spine and the chest is more limited in patients with scoliosis. The authors concluded that the general stiffness of the chest and the spine may contribute to the mechanical inefficiency and impairment of the pulmonary function observed in patients with scoliosis.

Studies comparing the outcomes of brace treatment or surgical treatment in boys versus girls revealed that the results of both were inferior among boys. The authors suggest that the treatment is less successful in boys due to relative stiffness of the spine and thorax and less compliance in brace wearing [7,9]. Furthermore, studies assessing the general joint hypermobility with use of the Beighton scale showed that the joint hypermobility prevalence is significantly lower in boys than in girls [36,37,38]. Czaprowski et al. [39] concluded that the prevalence of generalized joint hypermobility is significantly higher in patients with idiopathic scoliosis compared to the control group (in girls, *p* = 0.0054; in boys, *p* = 0.017).

Spine and chest deformation, followed by the impaired chest mechanics that influence the breathing pattern, may contribute to the restrictive pattern that is observed in patients with scoliosis [40]. Although, the restriction can be only diagnosed based on the body plethysmography results [28,41], regular spirometry examination allows the recognition of pulmonary impairment and indicates a possible need for further examination [42].

## 5. Conclusions

Pulmonary parameters tend to be lower in boys than in girls with severe thoracic idiopathic scoliosis. It may suggest the need for more intensive, dedicated physiotherapy in boys as a part of the preoperative preparation (prerehabilitation).

## Figures and Tables

**Table 1 healthcare-10-01574-t001:** Age, measured body height, corrected body height, and body weight in boys versus girls.

Parameter	Boysn = 30	Girlsn = 30	*p*-Value
Age (y.o.)	15.9 ± 1.3 (14–18)	15.6 ± 1.6 (14–18)	*p* = 0.437
Measured body height (cm)	172.8 ± 6.8 (158–185)	161.7 ± 5.5 (151–176)	*p* < 0.001 *
Corrected body height (cm)	176.7 ± 6.5 (162.1–188.5)	165.2 ± 5.6 (155.6–178.5)	*p* < 0.001 *
Body weight (kg)	61.8 ± 9.0 (45–91)	51.9 ± 6.9 (39–64)	*p* < 0.001 *

* difference statistically significant.

**Table 2 healthcare-10-01574-t002:** Calculations of the predicted values of the pulmonary parameters using the GLI 2012 recommendations [20] for boys (n = 30) versus girls (n = 30).

Parameter	FVC_boys_	FVC_girls_	*p*-Value	FEV1_boys_	FEV1_girls_	*p*-Value
Predicted value (L)	4.94 ± 0.5	3.76 ± 0.3	*p* < 0.001 *	4.22 ± 0.4	3.33 ± 0.3	*p* < 0.001 *
LLN (L)	4.01 ± 0.4	3.02 ± 0.2	*p* < 0.001 *	3.39 ± 0.5	2.68 ± 0.2	*p* < 0.001 *
ULN (L)	5.88 ± 0.6	4.52 ± 0.4	*p* < 0.001 *	5.03 ± 0.5	3.96 ± 0.3	*p* < 0.001 *

All values are presented as a mean ± standard deviation. FVC_boys_—forced vital capacity in boys; FVC_girls_—forced vital capacity in girls; FEV1_boys_—forced expiratory volume in 1 s in boys; FEV1_girls_—forced expiratory volume in 1 s in girls; LLN—lower limit of normal; ULN—upper limit of normal. * difference statistically significant.

**Table 3 healthcare-10-01574-t003:** The absolute FVC and FEV1 values obtained during spirometry examination in boys (n = 30) versus girls (n = 30).

Parameter	Boys	Girls	*p*-Value
FVC (L)	3.49 ± 0.8 (2.1–5.2)	2.73 ± 0.6 (1.6–3.8)	*p* < 0.001 *
FEV1 (L)	2.95 ± 0.7 (1.8–3.9)	2.52 ± 0.8 (1.4–5.5)	*p* = 0.036 *

All values are presented as a mean ± standard deviation, followed by the minimum and maximum in brackets. FVC—forced vital capacity; FEV1—forced expiratory volume in 1 s. * difference statistically significant.

**Table 4 healthcare-10-01574-t004:** Comparison of the spirometry examination expressed as the FVC and FEV1 z-score and as the percentage of the predicted values of the FVC and FEV1 for boys (n = 30) versus girls (n = 30).

		FVC			FEV1	
Parameter	FVC_boys_	FVC_girls_	*p*-Value	FEV1_boys_	FEV1_girls_	*p*-Value
z-score	−2.44 ± 1.4	−2.29 ± 1.6	0.701	−2.52 ± 1.4	−1.99 ± 1.8	0.262
%	70.24 ± 17.3	73.19 ± 13.3	0.525	70.35 ± 19.3	75.83 ± 22.7	0.414

Values are presented as a mean ± standard deviation. FVC_boys_—forced vital capacity body height in boys; FVC_girls_—forced vital capacity in girls; %—percentage of the predicted value of the pulmonary parameters; FEV1_boys_—forced expiratory volume in 1 s in boys; FEV1_girls_—forced expiratory volume in 1 s in girls.

## Data Availability

The data presented in the study are available on request from the corresponding author.

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
