# Peer review of "Pulmonary Parameters in Adolescents with Severe Thoracic Idiopathic Scoliosis: Comparison Girls versus Boys"

_healthcare, 2022, doi:10.3390/healthcare10081574_

Round 1

Reviewer 1 Report

Thank you for inviting me to review this manuscript on “Pulmonary Parameters in Adolescents with Severe Thoracic Idiopathic Scoliosis: Comparison Girls Versus Boys.”

Unfortunately, your paper is not considered worthy of acceptance. The reasons are as follows.

Table 4 is not a significant result.

The gender difference in the FVC z-scores does not exceed 0.5.

Therefore, it does not suggest the need for more intensive preoperative pulmonary rehabilitation in men.

Reviewer 2 Report

line 131-138 and 148-152 are the summary of the results so should be transfered to aproppriate part of the paper

Reviewer 3 Report

Gender medicine is currently very advanced, so this paper is of high interest.

For what concerns the material and method, certainly a Corrected Body Height Calculation is important, but as my advice it would be interesting to also evaluate the body mass index BMI. For the type of journal, but with a medical orientation, the paper can be accepted. My notes derive from an experience above all surgery in this field, as can also be understood from my paper (Pulmonary function after thoracoplasty in the surgical treatment of adolescent idiopathic scoliosisT Greggi, G Bakaloudis, I Fusaro, M Di Silvestre, F Lolli… - Clinical Spine Surgery, 2010 )which unfortunately (perhaps I have escaped?) I do not see mentioned in the references.

Sincerely

Round 2

Reviewer 1 Report

 I have no further comments.